# Endogenous itaconate is not required for particulate matter-induced NRF2 expression or inflammatory response

Kaitlyn A Sun[1], Yan Li[2], Angelo Y Meliton[1], Parker S Woods[1], Lucas M Kimmig[1], Rengül Cetin-Atalay[1], Robert B Hamanaka[1], Gökhan M Mutlu[1]*

[1]Department of Medicine, Section of Pulmonary and Critical Care Medicine, The University of Chicago, Chicago, United States; [2]Center for Research Bioinformatics, The University of Chicago, Chicago, United States

**Abstract** Particulate matter (PM) air pollution causes cardiopulmonary mortality via macrophage-driven lung inflammation; however, the mechanisms are incompletely understood. RNA-sequencing demonstrated *Acod1* (*Aconitate decarboxylase 1*) as one of the top genes induced by PM in macrophages. *Acod1* encodes a mitochondrial enzyme that produces itaconate, which has been shown to exert anti-inflammatory effects via NRF2 after LPS. Here, we demonstrate that PM induces Acod1 and itaconate, which reduced mitochondrial respiration via complex II inhibition. Using *Acod1*[-/-] mice, we found that Acod1/endogenous itaconate does not affect PM-induced inflammation or NRF2 activation in macrophages in vitro or in vivo. In contrast, exogenous cell permeable itaconate, 4-octyl itaconate (OI) attenuated PM-induced inflammation in macrophages. OI was sufficient to activate NRF2 in macrophages; however, NRF2 was not required for the anti-inflammatory effects of OI. We conclude that the effects of itaconate production on inflammation are stimulus-dependent, and that there are important differences between endogenous and exogenously-applied itaconate.

*For correspondence:
gmutlu@uchicago.edu

Competing interests: The authors declare that no competing interests exist.

## Introduction

Exposure to particulate matter (PM) air pollution is associated with significant morbidity and mortality and is one of the top preventable causes of death in the world (*McGlade and Landrigan, 2019*). The World Health Organization estimates that exposure to air pollution is responsible for 4.2 million premature deaths worldwide every year (*WHO, 2018*). Air pollution has been identified as a leading cause of global disease burden, and as the fifth mortality risk factor particularly in low- and middle-income countries (*Cohen et al., 2017*). The majority of PM-associated morbidity and mortality is due to cardiopulmonary diseases including asthma, chronic obstructive lung disease, lung cancer, congestive heart failure and ischemic/thrombotic cardiovascular disease (myocardial infarction, ischemic stroke) (*Cohen et al., 2017*; *Hamanaka and Mutlu, 2018*).

We have previously shown that lung macrophages are required for PM-induced lung inflammation and consequently acute thrombotic events (*Mutlu et al., 2007*; *Chiarella et al., 2014*; *Soberanes et al., 2019*). PM induces the release of pro-inflammatory cytokines including interleukin-6 (IL-6), which is required for the PM-induced prothrombotic state and resultant acceleration of vascular thrombosis (*Mutlu et al., 2007*; *Chiarella et al., 2014*). Furthermore, we found that PM affects mitochondrial function in lung macrophages characterized by increased oxygen consumption rate and generation of reactive oxygen species (ROS), which are both required for PM-induced IL-6 production (*Soberanes et al., 2019*).

Despite significant improvement in our understanding about the mechanisms by which PM induces pro-inflammatory cytokines such as IL-6, the mechanisms that regulate PM-induced lung

**eLife digest** Air pollution is a major global health problem that causes around 4.2 million deaths each year. Once inhaled, pollution particles can remain in the lungs, where they cause inflammation, tissue damage, and ultimately chronic disease. Macrophages, a population of immune cells in the lungs, are involved in this inflammatory process.

Itaconate is a molecule with potential anti-inflammatory effects, produced by mammalian cells including macrophages. Recent studies have shown that a modified form of the molecule, 4-octyl itaconate, reduces inflammation when applied to cells exposed to lipopolysaccharide, a component of infectious bacteria that is, usually, a strong trigger of inflammation. These experiments used the 4-octyl modification to ensure that itaconate could get into the cells.

Itaconate's anti-inflammatory action is thought to work by activating a signaling process in cells called the NRF2 pathway. NRF2 is a protein made by 'active' macrophages, that is, macrophages already primed to respond to foreign particles. NRF2 in turn increases production of factors that 'damp down' inflammation, all of which are collectively termed the NRF2 anti-inflammatory pathway. Although macrophages in the lungs are linked with inflammation caused by air pollution, their role – and that of itaconate – is still not well-understood. Sun et al. therefore wanted to determine if itaconate helps macrophages control pollution-induced inflammation.

Initial experiments treated mouse macrophage cells with pollution particles. Analyzing gene activity in these cells showed that exposure to pollution did indeed switch on the *Acod1* gene, which encodes the enzyme that makes itaconate. It also turned on genes for other molecules involved in inflammation. Pre-treating macrophages with 4-octyl itaconate before pollution exposure reduced inflammation and also, as expected, turned on the NRF2 pathway.

To determine whether cells' own production of itaconate affected lung inflammation, macrophages were isolated from mutant mice lacking *Acod1*. Comparing these cells, which could not make itaconate, with normal cells revealed that removing itaconate did not change the inflammatory response to pollution. Activity of the NRF2 pathway also remained similar in both types of cells. This showed that itaconate produced by macrophages likely has different effects on lung inflammation from other forms of the compound.

These findings represent a step forward in understanding how pollution interacts with immune cells in the lungs. They reveal that the source of anti-inflammatory factors can be just as important in shaping immune responses as the type of factor. These results highlight the need for further, detailed work on the mechanisms underlying pollution-induced disease.

inflammation are not completely understood. To better understand these mechanisms, we performed RNA-sequencing in macrophages following PM exposure. Here, we demonstrate that PM regulates the expression of genes that have not been previously reported. One of the top genes induced by PM was *Irg1* (Immune-responsive Gene 1) or *Acod1* (Aconitate decarboxylase 1), which encodes a mitochondrial enzyme that produces itaconate from the Tricarboxylic Acid (TCA) cycle metabolite cis-aconitate (*Strelko et al., 2011*; *Michelucci et al., 2013*). Itaconate has been reported to be produced in macrophages following LPS stimulation and to have anti-inflammatory effects (*Lampropoulou et al., 2016*; *Mills et al., 2018*). We confirmed that PM induces both mRNA and protein expression of Acod1 in macrophages, and increases intracellular and media levels of itaconate. Our metabolic analysis showed that itaconate inhibits mitochondrial complex II (succinate dehydrogenase) and that Acod1 is an important regulator of cellular respiration following PM exposure. Treatment of macrophages with a cell-permeable itaconate derivative, 4-octyl itaconate (OI) attenuated PM- and LPS-induced cytokine production. $Acod1^{-/-}$ cells, which lack endogenous itaconate production, exhibited exaggerated IL-1β production following LPS exposure; however, no effect on PM-induced inflammation was observed. As recently described, OI induced NRF2 protein and expression of its target genes such as *Nqo1* and *Hmox1* (*Mills et al., 2018*); however, we found that NRF2 was not required for the anti-inflammatory effects of OI. Furthermore, using $Acod1^{-/-}$ cells, we found that endogenous itaconate production is not required for NRF2 protein or its target gene expression following treatment with either PM or LPS. These results suggest that recently reported anti-inflammatory and NRF2-inducing effects of exogenous itaconate (OI) may not represent the

effects of endogenously-produced itaconate and therefore OI should not be used as a surrogate for Acod1 and endogenous itaconate. Collectively, our results suggest that stimulus and location of itaconate production play major roles in governing the effect of itaconate on inflammation in macrophages. Furthermore, in contrast to recent studies, we suggest that NRF2 is not a major regulator of the anti-inflammatory effects of itaconate.

## Results

### Particulate matter induces aconitate decarboxylase 1 (*Acod1*) and production of itaconate in macrophages

Previous studies that evaluated the effect of PM on macrophages have evaluated a limited number of regulated genes, primarily focusing on pro-inflammatory cytokines. To gain a non-biased understanding of the mechanisms by which PM induces lung inflammation, we treated BMDMs from C57BL/6 mice with PM or vehicle control for 24 hours and performed RNA-Seq to analyze PM-induced changes in the transcriptome. Differentially expressed gene (DEG) analysis revealed 370 unique genes significantly regulated by PM. Gene ontology analysis of RNA-seq data showed upregulation of pathways relating to inflammatory responses, metabolism, and cytokine stimulation

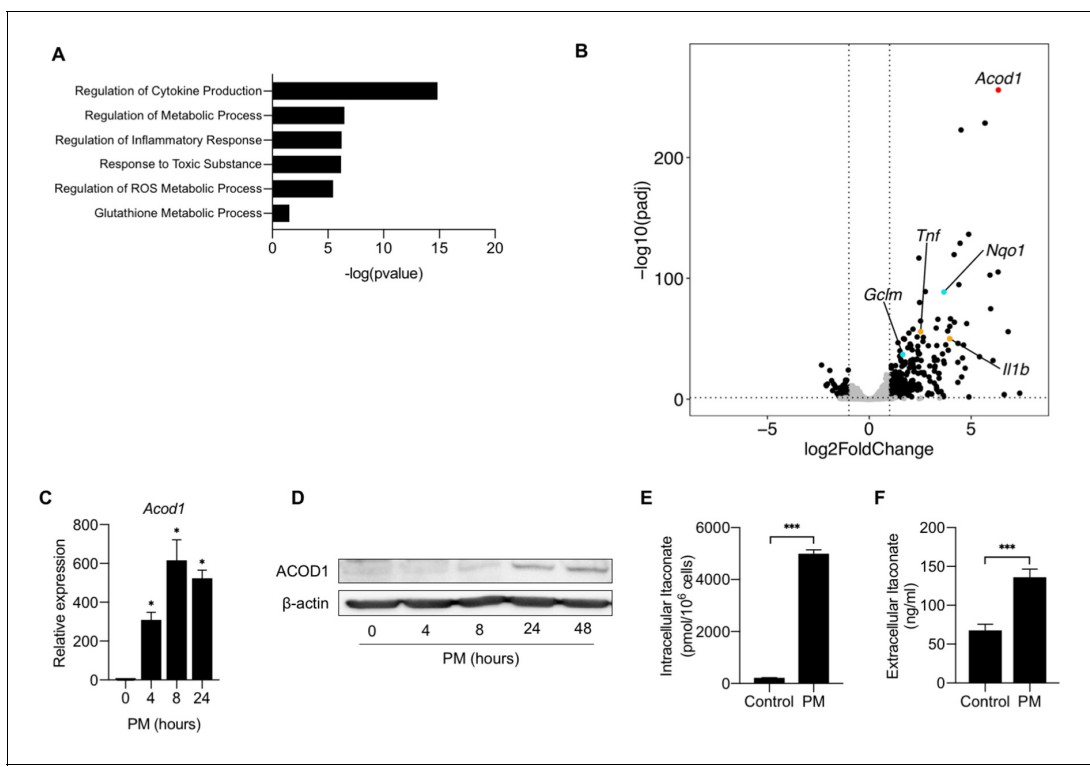

**Figure 1.** Aconitate Decarboxylase 1 (Acod1) is highly upregulated by particulate matter in macrophages. (**A**) We performed RNA-seq in BMDMs treated with either PM (20 μg/cm$^2$) or vehicle control for 24 hours. Gene ontology analysis highlight the pathways involved with PM treatment. (**B**) Volcano plot of RNA-seq data representing differentially expressed genes in BMDMs treated with PM or PBS (vehicle control) for 24 hours. *Acod1* (red point) is one of the most highly differentially expressed gene. In addition, we found induction of NRF2 target genes (i.e. *Nqo1, Gclm*; turquoise points) and inflammatory genes (i.e. *Tnf, Il1b*; orange points) following PM treatment. Black data points represent other significant genes, at FC > 2, and FDR adjusted p<0.05. (**C**) qPCR analysis of *Acod1* in BMDMs treated with PM for 4, 8 or 24 hours. Data are represented as fold change. Significance was analyzed with one-way ANOVA corrected with Bonferroni's post hoc test for multiple comparisons, *p<0.05, **p<0.01, ***p<0.001. (**D**) Western blot of ACOD1 protein at 0, 4, 8, 24 and 48 hours of PM treatment. (**E**) Intracellular itaconate concentration in BMDMs treated with PM for 24 hours, as measured by mass spectrometry. (**F**) Extracellular itaconate in media in BMDMs treated with PM for 24 hours, as measured using mass spectrometry. Significance was determined using two-tailed unpaired student's t-test, *p<0.05, **p<0.01, ***p<0.001.

The online version of this article includes the following source data for figure 1:

**Source data 1.** Differential expression analysis results of PM treated BMDMs compared with Control BMDMs.

(*Figure 1A*). The most highly upregulated gene induced by PM exposure was *Acod1* (*Figure 1B* and *Figure 1—source data 1*). Other highly upregulated genes included proinflammatory cytokine genes such as *Tnfa, Il1b* as well as *Nqo1,* a transcriptional target of Nuclear factor (erythroid-derived 2)-like 2 (NRF2) (*Figure 1B*).

*Acod1* encodes a mitochondrial enzyme, which catalyzes the conversion of cis-aconitate to itaconate, a mitochondrial metabolite that has recently been shown to regulate inflammatory responses (*Michelucci et al., 2013*). To confirm that PM upregulates mRNA and protein expression of Acod1, we first treated BMDMs with PM for 4, 8 or 24 hours, and then assessed the expression of *Acod1* through qPCR and western blot over time. PM induced *Acod1* gene expression as early as 4 hours after treatment, peaking at 8 hours (*Figure 1C*). Western blot analysis showed expression of ACOD1 protein was delayed compared to mRNA expression and was detectable beginning 24 hours after PM treatment (*Figure 1D*). To determine whether Acod1 protein induction was associated with increased cellular levels of its metabolic product, we measured intracellular levels of itaconate in BMDMs 24 hours after treatment with PM using capillary electrophoresis–mass spectrometry (CE-MS) (Human Metabolome Technologies, Boston, MA). We found that PM caused a significant increase in intracellular itaconate levels (*Figure 1E*), correlating with the upregulation of ACOD1. Furthermore, we detected increased concentrations of itaconate via mass spectrometry in the medium of macrophages treated with PM, indicating that itaconate is released from cells (*Figure 1F*). Collectively, these findings demonstrate that PM causes a time-dependent expression of ACOD1 and production of itaconate, prompting us to further study whether itaconate plays a role in the regulation of PM-induced inflammation.

## Itaconate decreases mitochondrial oxygen consumption by inhibiting succinate dehydrogenase/complex II in macrophages

Since itaconate has been shown to be a weak inhibitor of complex II, succinate dehydrogenase (SDH) (*Cordes et al., 2016*; *Lampropoulou et al., 2016*), we hypothesized that PM exposure may regulate mitochondrial respiration in macrophages through induction of Acod1 and production of itaconate.

We first confirmed that itaconate inhibits SDH in BMDMs using the Seahorse XF Plasma Membrane Permeabilizer to assess the effect of itaconate on individual mitochondrial respiratory complexes. To measure the effect of itaconate on complex II/SDH activity, we permeabilized cells in the presence of rotenone (to eliminate the contribution of complex I), ADP, and succinate as substrate. Measurement of oxygen consumption rate (OCR) showed an immediate decrease in OCR following injection of either itaconate or malonate, a known SDH inhibitor used as a positive control. Addition of oligomycin did not cause a further reduction in OCR (*Figure 2A*). These data suggest that itaconate, like malonate, is indeed an SDH inhibitor. When cells were permeabilized in media containing only the complex I substrates pyruvate/malate, injection of itaconate caused a smaller decrease in OCR relative to that seen during complex II-dependent respiration. This reduction was again similar to what was observed with malonate (complex II inhibitor). Injection of oligomycin resulted in a significant drop in OCR. These results are consistent with itaconate mediating its inhibitory effects on respiration via complex II (*Figure 2B*).

Because there are no known mechanisms of transporting itaconate into the cell, studies have used different forms of itaconate that are cell membrane-permeable to examine the role of itaconate in vitro. 4-octyl itaconate (OI) has been shown to cross the plasma membrane and increase intracellular concentrations of itaconate (*Mills et al., 2018*). We thus measured OCR in intact cells in the presence or absence of OI (0.25 mM, a dose shown by Mills et al to reduce LPS-induced inflammation [*Mills et al., 2018*]). OI injection caused a small reduction in basal OCR, while basal ECAR increased slightly, which is likely as a compensatory response to reduction in OCR (*Figure 2C and D*). Oligomycin then decreased the OCR of both groups to the same level consistent with OI-induced reduction in coupled respiration. Maximal OCR determined following FCCP, an uncoupler, was lower in OI-treated BMDMs compared to control BMDMs (*Figure 2E*). Together, these results suggest that itaconate is sufficient to reduce mitochondrial OCR via inhibition of complex II/SDH.

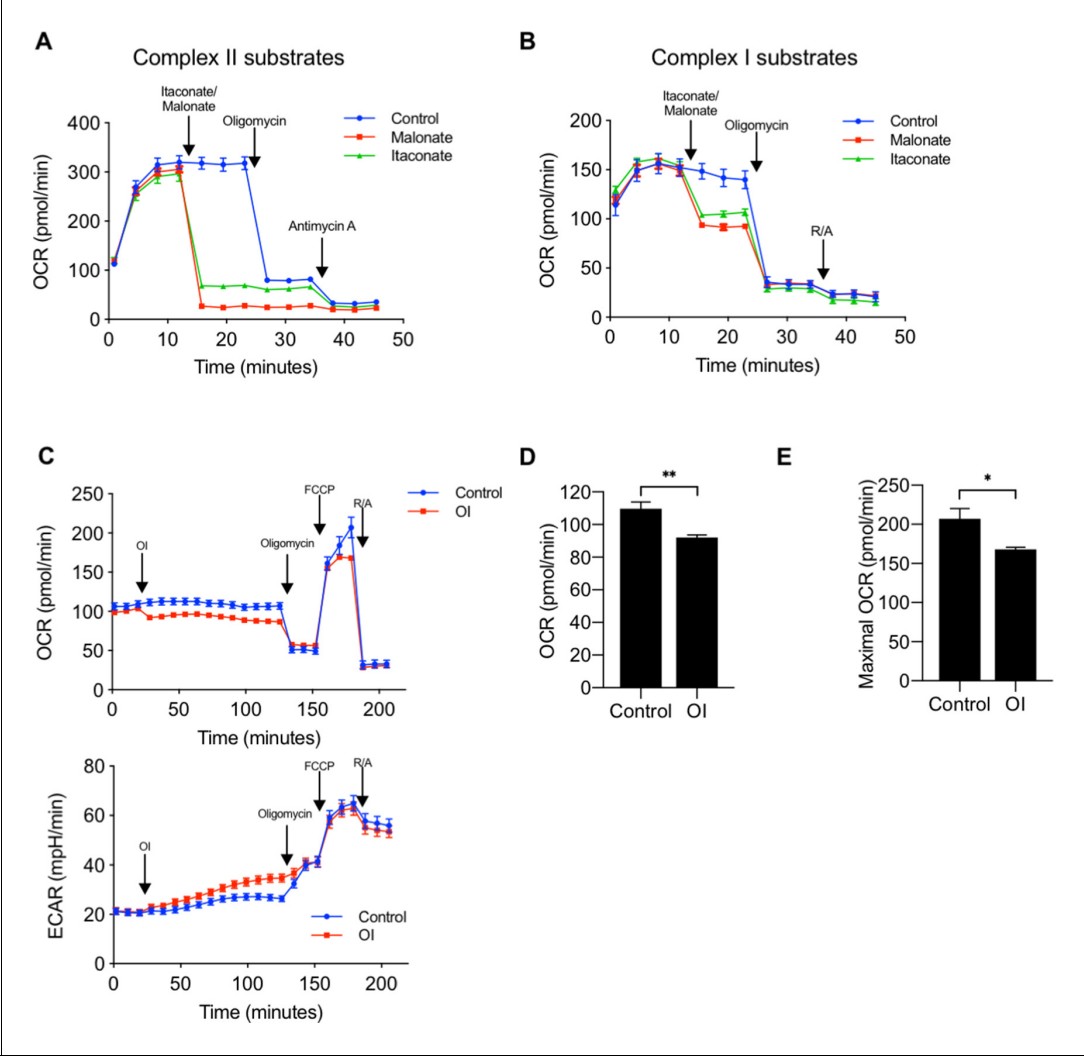

**Figure 2.** Itaconate decreases oxygen consumption rate via inhibition of complex II, succinate dehydrogenase (SDH). (A) We measured oxygen consumption rate (OCR) in permeabilized BMDMs (using XF plasma membrane permeabilizer) in the presence of ETC complex II substrate (succinate, 10 mM) and complex I inhibitor (rotenone, 2 mM), followed by injections of 1) media (control, $n$ = 6), itaconate (10 mM, $n$ = 6) or malonate (10 mM, $n$ = 6), a known complex II inhibitor, 2) oligomycin (2 mM), an ATP synthase inhibitor, and 3) Antimycin A (2 mM), a complex III inhibitor. Both itaconate and malonate significantly decreased OCR. (B) We measured OCR in permeabilized BMDMs in the presence of ETC complex I substrates (pyruvate, 10 mM and malate, 0.5 mM) followed by injections of 1) itaconate or malonate, 2) oligomycin, or 3) rotenone and antimycin A (R/A) ($n$ = 6). (C) We performed a mitochondrial stress test to measure OCR and ECAR in BMDMs treated with 4-octyl itaconate (OI) (0.25 mM) or vehicle control ($n$ = 8). OI acutely decreased both (D) basal OCR and (E) maximal OCR measured at third time point after FCCP. Significance was determined using two-tailed unpaired student's t-test, *p<0.05, **p<0.01, ***p<0.001.

## PM causes time-dependent effects on mitochondrial oxygen consumption: An initial increase in OCR is followed by a late reduction in OCR

Since itaconate inhibits mitochondrial respiration and ACOD1 is induced late following PM exposure, we sought to determine whether PM exposure exerts time-dependent changes in mitochondrial oxygen consumption rate in macrophages. We thus measured OCR in BMDMs following PM treatment at 1 hour and 24 hours, when ACOD1 expression is undetectable and present, respectively. As we have previously shown (*Soberanes et al., 2019*), after 1 hour of exposure, cells treated with PM exhibited increased basal oxygen consumption rate compared with control BMDMs (*Figure 3A*). Consistent with a role of glycolysis in promoting inflammation, ECAR was also induced by PM exposure after 1 hour. Interestingly, after 24 hours of PM treatment, ECAR levels remained elevated;

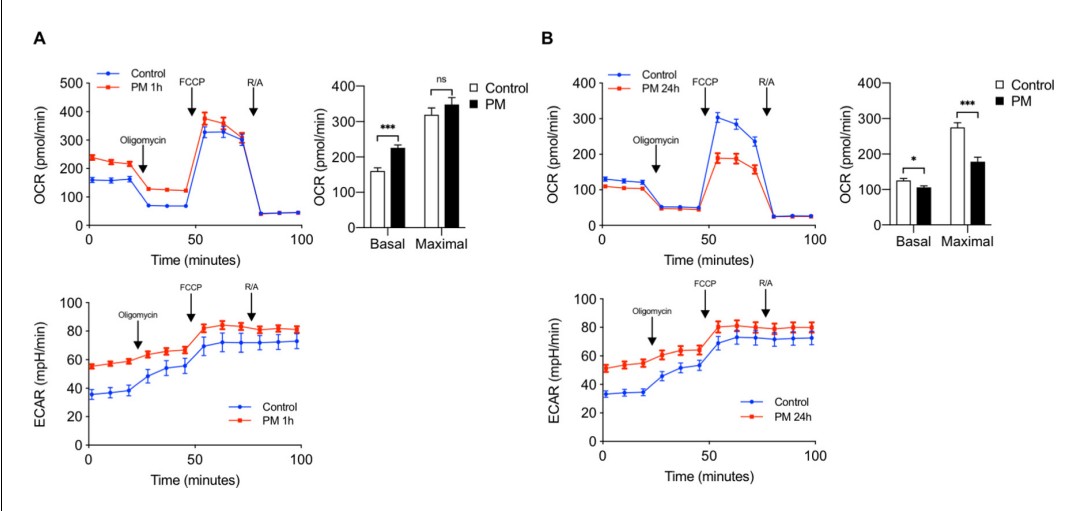

**Figure 3.** The effect of PM on mitochondrial oxygen consumption is time dependent. (**A**) We performed a mitochondrial stress test to measure OCR and ECAR in BMDMs at (**A**) 1 hour or (**B**) 24 hours following treatment with PM (20 μg/cm$^2$) or control vehicle control. Oligomycin (ATP synthase inhibitor), FCCP (uncoupler), and rotenone/antimycin A (R/A) (complex I/III inhibitors) were injected sequentially. Basal OCR was acutely increased at 1 hour, which is before the induction of ACOD1 protein and production of itaconate. In contrast, both basal and maximal OCR decreased at 24 hours after PM treatment, when ACOD1 protein and itaconate levels are high (n = 8). ECAR was increased with PM at both timepoints. Significance was determined using two-tailed unpaired student's t-test, *p<0.05, **p<0.01, ***p<0.001.

however, both basal and maximal OCR were decreased relative to control-treated cells. (*Figure 3B*). This decrease in OCR following 24 hours of PM treatment was similar to the decrease in OCR after itaconate treatment (*Figure 2C*) suggesting that ACOD1 induction and itaconate production are important regulators of macrophage metabolism following PM treatment.

## Endogenous itaconate production is required for PM-induced reduction in mitochondrial respiration and metabolic reprogramming in macrophages

We next sought to determine whether PM-induced ACOD1 expression and endogenous itaconate production is required for PM-induced changes in mitochondrial metabolism at 24 hours. In order to answer this question, we used BMDMs from *Acod1*$^{-/-}$ mice, which lack the ability to produce itaconate. We first measured TCA cycle intermediates in WT and *Acod1*$^{-/-}$ BMDMs at 24 hours following treatment with PM or vehicle. PM-induced production of itaconate was detectable only in WT but not in *Acod1*$^{-/-}$ BMDMs, confirming that ACOD1 is required for itaconate production (*Figure 4A,B*). Consistent with the role of itaconate as an inhibitor of SDH, succinate accumulated in WT BMDMs, but not *Acod1*$^{-/-}$ cells following PM treatment (*Figure 4A,C*). This finding provides further support for endogenously-produced itaconate inhibiting SDH and leading to accumulation of succinate, as SDH catalyzes the oxidation of succinate to fumarate. In contrast, *Acod1*$^{-/-}$ BMDMs had higher levels of TCA metabolites downstream of succinate (fumarate, malate) (*Figure 4A*), consistent with the idea that due to the absence of itaconate in these cells, SDH and consequently the oxidation of succinate to fumarate are not inhibited. These results are consistent with previous reports using LPS as a stimulus (*Lampropoulou et al., 2016*). Overall, our metabolomics data suggest that PM causes a break in the TCA cycle that is Acod1 dependent.

To determine whether PM-induced ACOD1 expression and production of itaconate are required for the reduction of OCR at 24 hours following PM, we performed a mitochondrial stress test in WT and *Acod1*$^{-/-}$ BMDMs. While WT BMDMs exhibited significantly reduced OCR following PM, *Acod1*$^{-/-}$ cells treated with PM for 24 hours did not show any reduction in OCR compared to control treatment (*Figure 4D*). Collectively, these results suggest that endogenously-produced itaconate is required for PM-induced reduction in OCR.

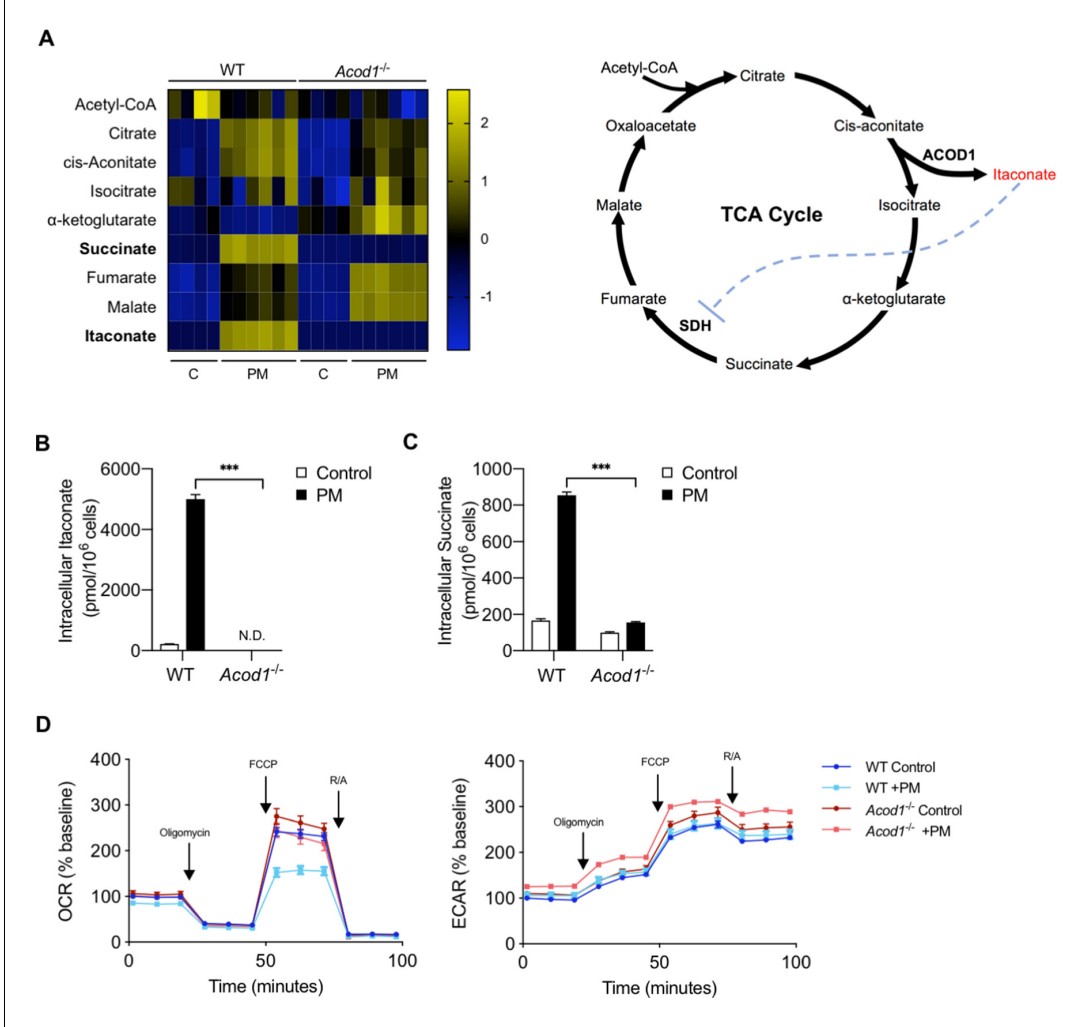

**Figure 4.** Acod1 and endogenous itaconate production is required for PM-reduced mitochondrial OCR via inhibition of SDH. (**A**) Heatmap of intracellular levels of TCA cycle metabolites and itaconate at 24 hours following treatment with PM. (**B–C**) Intracellular concentrations of (**B**) itaconate and (**C**) succinate in WT and *Acod1*-/- cells with (*n* = 6) and without (*n* = 4) PM; itaconate is not detectable in *Acod1*-/- cells, while succinate does not accumulate in *Acod1*-/- cells. (**D**) Mitochondrial stress test of WT and *Acod1*-/- BMDMs after 24 hours of PM treatment. OCR was normalized to baseline OCR in WT control BMDMs (untreated). Maximal OCR levels are only decreased in WT cells, and not *Acod1*-/- cells following PM. Significance was analyzed with one-way ANOVA corrected with Bonferroni's post hoc test for multiple comparisons, *p<0.05, **p<0.01, ***p<0.001.
The online version of this article includes the following figure supplement(s) for figure 4:

**Figure supplement 1.** Inducible Nitric Oxide Synthase (iNOS) is induced by LPS but no by PM.

While a recent study showed that nitric oxide production as a result of LPS exposure can modulate metabolic reprogramming and affect oxidative phosphorylation in macrophages (*Bailey et al., 2019*), we found that inducible nitric oxide synthase (iNOS) protein was produced only in BMDMs stimulated with LPS. PM did not induce iNOS expression in either wild-type or *Acod1*-/- BMDMs (*Figure 4—figure supplement 1*). Thus, nitric oxide production is likely to be minimal with PM as a stimulus, and would not play a large role in the regulation of oxidative phosphorylation following PM.

## Exogenous, but not endogenous itaconate attenuates the PM-induced inflammatory response in macrophages

We have previously shown that PM induces an inflammatory response in macrophages, including the release of proinflammatory cytokines such as IL-6 and TNFα (*Mutlu et al., 2007*; *Chiarella et al., 2014*). Since the expression of ACOD1 was time-dependent, we first sought to determine whether

the effect of PM on cytokine expression was also time-dependent. We thus treated BMDMs with PM for 4, 8 or 24 hours and then analyzed the pro-inflammatory cytokine mRNA expression. We found that the mRNA expression of *Il6, Tnfa* and *Il1b* increased at 4 and 8 hours, and then decreased at 24 hours suggesting that PM-induced expression of cytokines is time-dependent similar to the expression of ACOD1 (*Figure 5A*).

Since the reduction in inflammatory cytokine mRNA coincided with the induction of ACOD1 and itaconate production, we hypothesized that itaconate may play a role in the reduction in cytokine expression observed at 24 hours. To test this hypothesis, we first evaluated the effect of exogenous itaconate on the early PM-induced inflammatory response. We pre-treated BMDMs with OI for

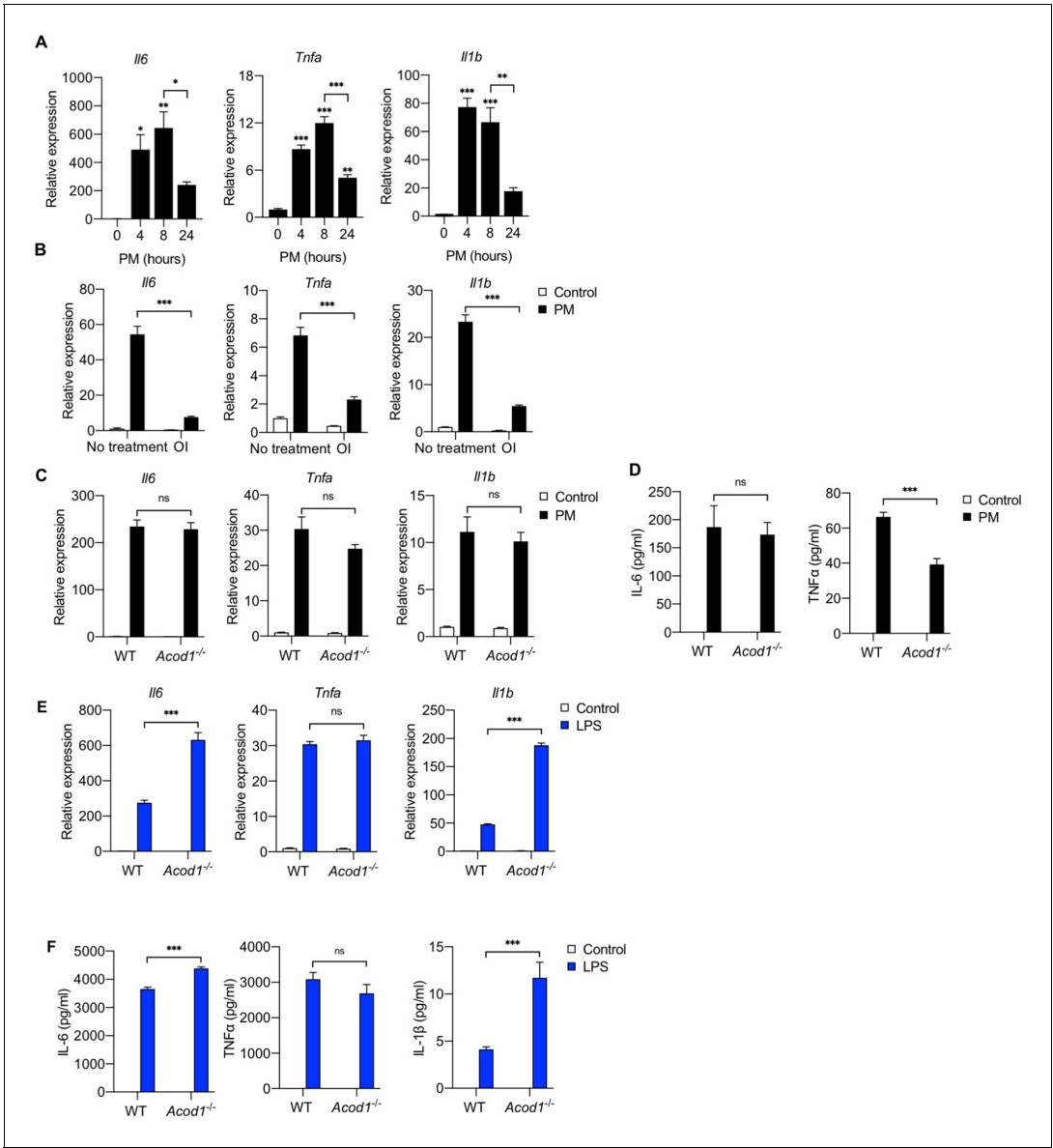

**Figure 5.** Exogenous but not endogenous itaconate decreases PM-induced inflammation. (**A**) We treated WT BMDMs with PM for 4, 8 or 24 hours and measured mRNA expression of *Tnfa, Il6* and *Il1b* by qPCR (*n* = 3). (**B**) We pretreated WT BMDMs with OI or vehicle control (DMSO) for 2 hours before we treated them with PM or vehicle control for 4 hours. We then measured mRNA expression of *Tnfa, Il6* and *Il1b* (qPCR). (**C, D**) We treated WT and *Acod1⁻/⁻* BMDMs with PM or ve and measured (**C**) mRNA expression of *Tnfa, Il6* and *Il1b* (qPCR). and (**D**) protein levels of IL-6 and TNFα in media (ELISA). IL-1β protein was not detectable. (**E, F**) We treated WT and *Acod1⁻/⁻* BMDMs with LPS (100 ng/ml) or PBS (control) and measured (**E**) mRNA expression of *Tnfa, Il6* and *Il1b* (qPCR). and (**F**) protein levels of IL-6, TNFα, and IL-1β in media (ELISA, *n* = 4). Significance was analyzed with one-way ANOVA corrected with Bonferroni's post hoc test for multiple comparisons *p<0.05, **p<0.01, ***p<0.001.

2 hours before treating with PM for 4 hours and measured mRNA expression of pro-inflammatory cytokines by qPCR. Pretreatment with OI attenuated the PM-induced mRNA expression of *Tnfa*, *Il6* and *Il1b*, suggesting that accumulation of itaconate with the induction of ACOD1 may be responsible for reduced cytokine mRNA expression at 24 hours following PM treatment (*Figure 5B*).

To determine whether absence of itaconate accumulation would exaggerate PM-induced inflammation at later timepoints, we measured cytokine mRNA and protein level in both wild-type and *Acod1⁻/⁻* BMDMs 24 hours after exposure to PM. Interestingly, loss of itaconate accumulation did not result in increased mRNA expression of *Il6*, *Tnfa* or *Il1b* or protein level in media in *Acod1⁻/⁻* BMDMs (*Figure 5C,D*). While loss of Acod1/itaconate did not affect PM-induced IL-6 protein levels, there was a reduction in TNFα protein level in media (*Figure 5D*). Overall, these results suggest that endogenous itaconate does not attenuate the PM-induced inflammatory response.

This was a surprising finding as Acod1/itaconate deficiency has been shown to augment inflammatory cytokine expression in response to LPS (*Lampropoulou et al., 2016*). To ensure that our findings were not the result of experimental differences other than stimulus, we treated wild-type and *Acod1⁻/⁻* BMDMs with LPS for 24 hours and measured *Il6*, *Tnfa* or *Il1b* mRNA expression. In contrast with PM treatment, *Acod1⁻/⁻* cells exhibited augmented *IL6* and *Il1b* mRNA expression after LPS treatment (*Figure 5E*), consistent with previously published results (*Lampropoulou et al., 2016*). There was no effect of *Acod1* deletion on LPS-induced *Tnfa* expression (*Figure 5E*), also consistent with previous findings (*Lampropoulou et al., 2016*). The loss of Acod1/itaconate also resulted in increased IL-6 and IL-1β protein level in media (*Figure 5F*). These findings suggest that the effect of itaconate on inflammation is stimulus-dependent. Importantly, these results also suggest that the effect of endogenously-produced itaconate on inflammation is different than that of exogenously-applied itaconate.

## Differential effects of endogenous versus exogenous itaconate on the PM-induced transcriptomic response

To better understand the role of endogenous and exogenously-applied itaconate on PM-induced response in macrophages, we performed RNA-sequencing in *Acod1⁻/⁻* and WT BMDMs exposed to PM (*Figure 6A* and *Figure 6—source data 1*). We found only 51 DEGs between *Acod1⁻/⁻* and WT BMDMs following PM (*Figure 6B* and *Figure 6—source data 2*). Both inflammatory genes, including *Il1b* and *Tnfa*, and NRF2 target genes, including *Nqo1* and *Gclm*, were not significantly different between *Acod1⁻/⁻* and WT BMDMs. Next, to determine the effect of exogenous itaconate on PM-induced gene expression, we performed RNA-sequencing in WT BMDMs exposed to PM following 2 hours of pretreatment with vehicle or OI. Transcriptomic analysis showed 1,030 DEGs between groups with and without OI (*Figure 6C* and *Figure 6—source data 3*). OI pretreatment significantly downregulated inflammatory gene expression in PM-treated BMDMs and upregulated NRF2 target genes. The effect of OI on these genes was not significantly different between WT and *Acod1⁻/⁻* cells. These results suggest that the effect of exogenous itaconate on the PM-induced transcriptomic response is markedly different from the effect of endogenous itaconate (*Figure 6A*).

## Endogenous itaconate production is not required for PM- or LPS-induced NRF2 expression

Recent studies suggest that itaconate exerts its inhibitory effects on LPS-induced inflammation via activation of NRF2, a transcriptional factor that plays a key role in antioxidant defense (*Mills et al., 2018*). Both PM and LPS induce mitochondrial production of reactive oxygen species which promote inflammatory gene expression (*Hsu and Wen, 2002*; *Soberanes et al., 2012*); however, PM contains a mixture of metals and other compounds which have their own redox-modulating properties (*Jeng, 2010*). We thus hypothesized that differential effects of PM and LPS on NRF2 activation may explain the lack of effect of endogenous itaconate on inflammation in PM-treated *Acod1⁻/⁻* BMDMs. We found that PM upregulated NRF2 and its target genes (*Nqo1*, *Gclm*, *Hmox1*) equally in WT and *Acod1⁻/⁻* BMDM (*Figure 7A,B*). These results suggested that itaconate-independent activation of NRF2 might explain the fact that no augmented inflammatory gene expression was observed in *Acod1⁻/⁻* cells after PM exposure.

As our experiments in *Acod1⁻/⁻* BMDMs showed that Acod1 is not required for PM-induced NRF2 activation, we sought to determine whether endogenous itaconate production is required for LPS-

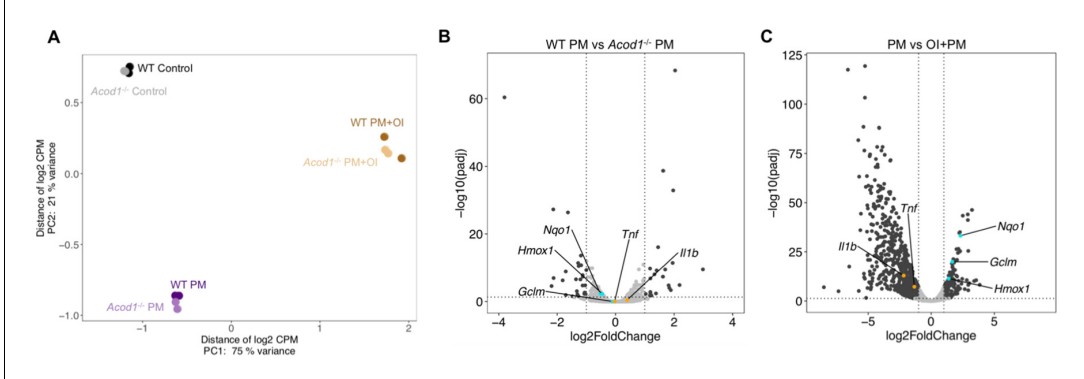

**Figure 6.** Endogenous and exogenous Itaconate have different effects on transcriptomic changes in response to PM. (**A**) PCA plot showing top 500 of 10,250 low expression removed gene features in WT or *Acod1*[-/-] BMDMs treated with PM and/or OI for 24 hours. (**B–C**) Volcano plots showing differentially expressed genes (DEGs) in (**B**) PM-treated *Acod1*[-/-] BMDMs compared with PM-treated WT BMDMs (51 DEGs), and (**C**) OI and PM-treated WT BMDMs compared with only PM-treated WT BMDMs (1,030 DEGs). DEGs were identified using DESeq2 at FC > 2 and FDR adjusted p-value of p<0.05. Dark gray points represent significantly different genes; light gray points represent not significantly different genes. Inflammatory genes (orange) and NRF2 target genes (turquoise) were not significantly different between WT and *Acod1*[-/-] BMDMs, while OI-pretreated BMDMs significantly expressed fewer inflammatory genes and more NRF2 target genes compared to BMDMs without OI pretreatment.

The online version of this article includes the following source data for figure 6:

**Source data 1.** Table of gene feature counts for all samples as generated using featureCounts.
**Source data 2.** Differential expression analysis results of PM treated *Acod1*[-/-] BMDMs compared with PM treated WT BMDMs.
**Source data 3.** Differential expression analysis results of OI and PM treated BMDMs compared with only PM treated BMDMs.

induced NRF2 activation. Surprisingly, we found that similar to PM-induced NRF2 activation, Acod1 expression was not required for LPS-mediated induction of NRF2 protein or target gene expression (*Figure 7C,D*). NRF2 protein levels were similarly upregulated in both LPS-treated WT and *Acod1*[-/-] cells (*Figure 7C*). While *Gclm* expression was slightly reduced in *Acod1*[-/-] cells, *Hmox1* induction was not affected by *Acod1* deficiency and *Nqo1* was more highly induced by LPS in *Acod1*[-/-] cells (*Figure 7D*).

Furthermore, we found that NRF2 induction after LPS treatment occurs prior to Acod1 induction (*Figure 7E*). Treatment with LPS induced a time-dependent expression of ACOD1 and NRF2. Interestingly, expression of NRF2 occurred at an earlier time point compared to ACOD1 (4 hours vs. 8 hours). Maximal expression of NRF2 also preceded the maximal expression of ACOD1 (8 hours vs 24 hours). Taken together, our results suggest that although NRF2 activation has been proposed to be the mechanism by which itaconate exerts its inflammatory effect, NRF2 induction occurs prior to ACOD1, and furthermore, ACOD1 is not required for NRF2 activation downstream of either PM or LPS.

## NRF2 is not required for the anti-inflammatory effects of exogenous itaconate

To date, the majority of studies investigating the role of Acod1 and itaconate on inflammation, including those linking NRF2 to the mechanisms by which itaconate may regulate inflammation, have largely focused on the effects of exogenously-applied cell membrane-permeable forms of itaconate (e.g., dimethyl itaconate and OI) (*Lampropoulou et al., 2016*; *Bambouskova et al., 2018*; *Mills et al., 2018*; *Zhao et al., 2019*). NRF2 has been suggested to regulate the effect of OI on IL-1β expression; however, as we saw that OI reduced PM and LPS-induced expression of other cytokines, including TNFα, the expression of which is not regulated by endogenous itaconate, we examined whether the effect of OI on these cytokines was regulated by NRF2. We found that, consistent with previous findings (*Mills et al., 2018*), OI treatment was sufficient to induce NRF2 protein and expression of NRF2 target genes (*Nqo1*, *Gclm*, and *Hmox1*) (*Figure 8A,B*). OI induced NRF2 protein expression to a similar extent as PM, and the combination of OI and PM caused a further increase in NRF2 protein levels and expression of its target gene, *Nqo1* (*Figure 8A,B*).

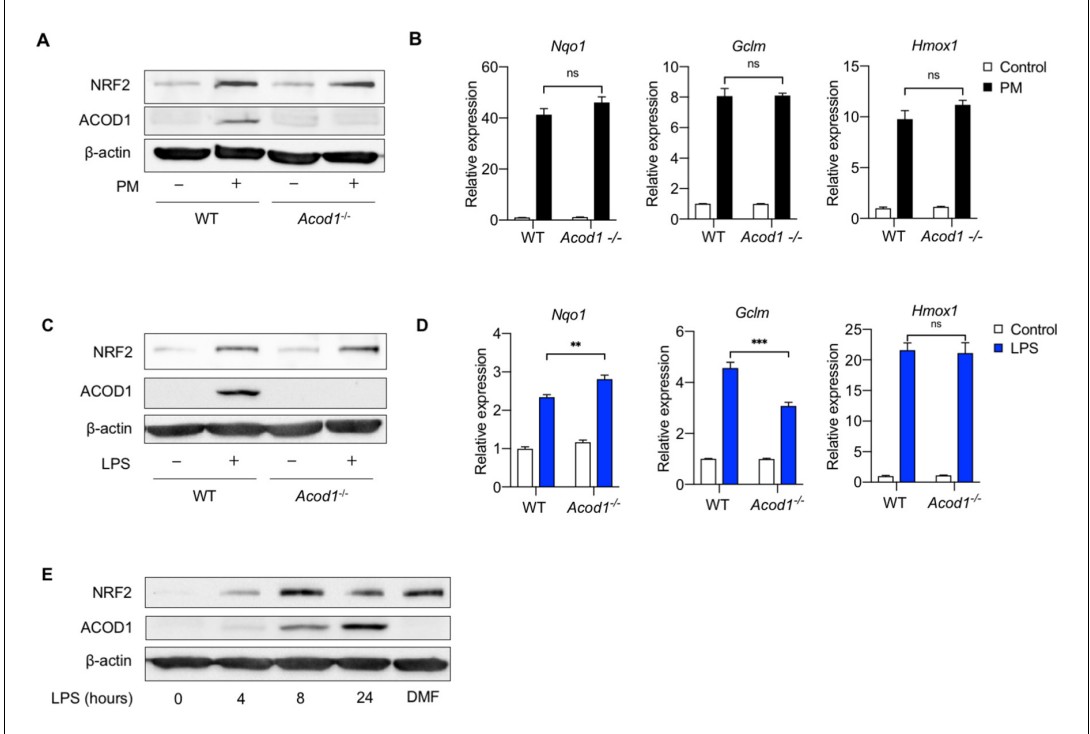

**Figure 7.** Acod1 and endogenous Itaconate production is not required for PM and LPS-induced activation of NRF2 pathway. (A, B) We treated WT and *Acod1*^-/- BMDMs with PM for 24 hours and measured (A) protein expression of NRF2 (Western blot) and (B) mRNA expression of NRF2 target genes *Nqo1, Hmox1* and *Gclm* (qPCR). NRF2 protein and target genes are unchanged between WT and *Acod1*^-/- (C, D) We treated WT and *Acod1*^-/- BMDMs with LPS (100 ng/ml) for 24 hours and measured (C) protein expression of NRF2 (Western blot) and (D) mRNA expression of NRF2 target genes *Nqo1, Hmox1* and *Gclm* (qPCR). NRF2 protein is not different between WT and *Acod1*^-/- cells. (E) We treated WT BMDMs with LPS for 4, 8, and 24 hours and measured protein expression of NRF2 and ACOD1 (Western blot) over time. Dimethyl fumarate (DMF, 0.1 mM) was used as a positive control for NRF2 expression. The expression of NRF2 precedes ACOD1 and the peak expression of NRF2 occurs before the peak expression of ACOD1. Significance of qPCR data was analyzed with one-way ANOVA corrected with Bonferroni's post hoc test for multiple comparisons, *$p<0.05$, **$p<0.01$, ***$p<0.001$.

To determine whether NRF2 is required for the OI-mediated attenuation of inflammatory cytokine expression, we transfected BMDMs with two independent siRNAs targeting *Nfe2l2*, or a non-targeting control. NRF2 protein was confirmed to be eliminated by Western blot analysis, as *Nfe2l2* siRNA-transfected cells did not upregulate NRF2 protein following 4 hours of PM treatment (*Figure 8C*, *Figure 8—figure supplement 1*).

Despite the loss of NRF2, we found that OI still exhibited its anti-inflammatory effects on PM-induced inflammation. OI attenuated PM-induced proinflammatory cytokine (*Tnfa, Il6, Il1b*) expression in both control and NRF2 knockdown cells (*Figure 8D*). These data suggest that NRF2 is not required for exogenous itaconate to attenuate PM-induced inflammation and are in contrast to recent results suggesting that OI exerts its anti-inflammatory effect on LPS-induced inflammation by upregulating NRF2 (*Mills et al., 2018*).

Because of the differences between PM and LPS, we then looked at the effect of OI on LPS-induced inflammation. While OI similarly decreased LPS-induced inflammation (*Figure 9A*), in contrast to PM treatment, LPS alone did not induce significant NRF2 target gene activation after 4 hours. No additional effect was seen on NRF2 target gene expression with the combination of LPS and OI (*Figure 9B*). Furthermore, consistent with our findings with PM, OI reduced inflammatory gene expression in response to LPS independent of NRF2 induction (*Figure 9C*, *Figure 9—figure supplement 1*). Taken together, our results suggest that although NRF2 is induced by OI, this activation is not required for the anti-inflammatory effect of exogenous itaconate.

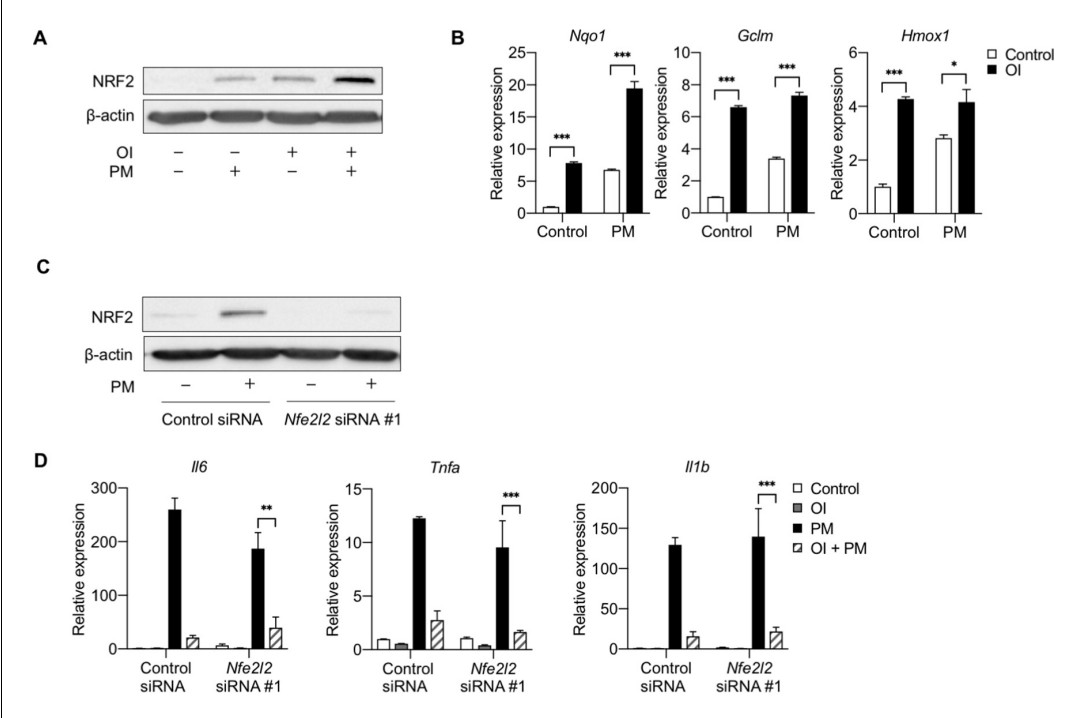

**Figure 8.** NRF2 is not required for the anti-inflammatory effects of exogenous itaconate (OI) on PM-induced inflammatory response. (**A**) Western blot showing upregulation of NRF2 protein following 2 hours of OI pretreatment (0.25 mM) followed by 4 hours of PM (20 µg/cm$^2$) treatment. The combination of OI and PM further increased NRF2 activation. (**B**) qPCR of NRF2 target genes (*Nqo1*, *Gclm*, and *Hmox1*) in WT BMDMs treated with PM for 4 hours, with or without OI pretreatment (0.25 mM, for 2 hours). (**C**) BMDMs transfected with scramble control siRNA or *Nfe2l2* siRNA (#1), Western blot analysis of control siRNA and *Nfe2l2* siRNA (#1)-transfected BMDMs following 4 hours of PM treatment to induce NRF2 protein. (**D**) qPCR of pro-inflammatory cytokine genes (*Tnfa*, *Il6* and *Il1b*) in *Nfe2l2* siRNA-transfected BMDMs treated with OI (2 hours of pretreatment) and PM (4 hours). Significance of qPCR was analyzed with one-way ANOVA corrected with Bonferroni's post hoc test for multiple comparisons, *p<0.05, **p<0.01, ***p<0.001.

The online version of this article includes the following figure supplement(s) for figure 8:

**Figure supplement 1.** NRF2 is not required for the anti-inflammatory effects of exogenous itaconate (OI) on PM-induced inflammatory response.

## Itaconate does not modulate PM-induced inflammation or NRF2 response in vivo

Lungs are the primary site of entry for PM, and we have previously demonstrated that alveolar macrophages (AMs) are metabolically distinct from BMDMs (*Woods et al., 2020*). We thus sought to determine whether the role of itaconate in AMs is similar to what we observed in BMDMs. We first isolated alveolar macrophages from WT and *Acod1*[-/-] mice and then treated them with PM as we did in BMDMs (*Figure 10—figure supplement 1*). As expected, there was no *Acod1* expression in *Acod1*[-/-] alveolar macrophages and treatment with PM induced a significant increase in the expression of *Acod1* mRNA in WT alveolar macrophages (*Figure 10—figure supplement 1A*). PM increased the expression of inflammatory cytokine genes (*Il6*, *Tnfa*, and *Il1b*) as well as IL-6 and TNFα protein levels in media both in WT and *Acod1*[-/-] alveolar macrophages (*Figure 10—figure supplement 1B,C*). There was no difference in cytokine mRNA or protein expression between WT and *Acod1*[-/-] alveolar macrophages. As we observed in BMDMs, PM caused similar increases in the expression of NRF2 target genes in WT and *Acod1*[-/-] alveolar macrophages (*Figure 10—figure supplement 1D*).

We next investigated whether itaconate plays a role in inflammation in an in vivo model. We intratracheally instilled either PBS or PM (in PBS) into the lungs of WT and *Acod1*[-/-] mice as we previously described (*Mutlu et al., 2006*; *Mutlu et al., 2007*). Twenty-four hours after instillation, we obtained bronchoalveolar lavage (BAL) fluid to measure cytokine levels as well as to isolate AMs for gene expression analysis. AM expression of *Acod1* was significantly induced by PM in WT mice only

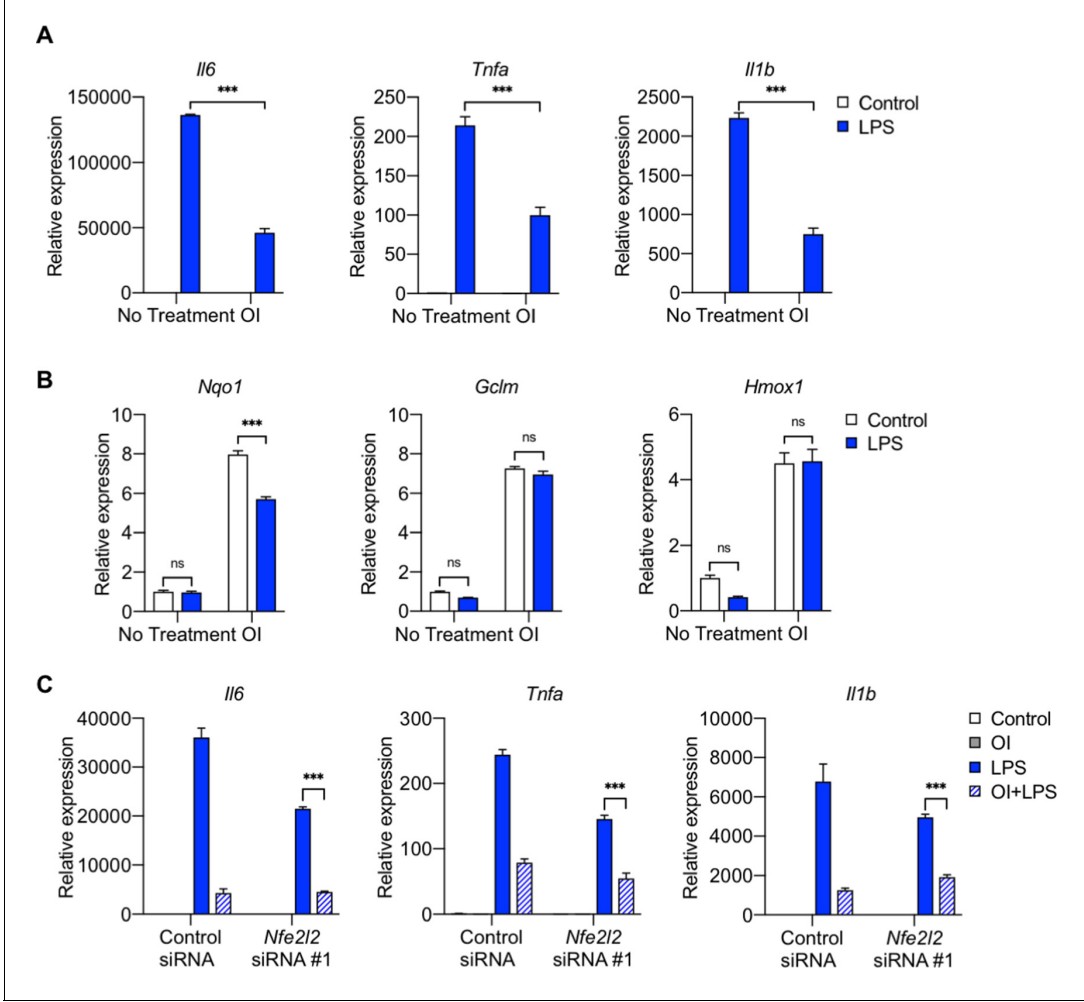

**Figure 9.** NRF2 is not required for the anti-inflammatory response of exogenous itaconate (OI) on LPS-induced inflammatory response. (A–B) qPCR of (A) proinflammatory cytokine (*Tnfa*, *Il6* and *Il1b*) genes and (B) NRF2 target genes (*Nqo1*, *Gclm*, and *Hmox1)* in WT BMDMs treated with LPS (100 ng/ml, 4 hours), with or without OI pretreatment (0.25 mM, 2 hours). (C) qPCR of pro-inflammatory cytokine genes (*Tnfa*, *Il6* and *Il1b*) in control or *Nfe2l2* siRNA (#1)-transfected BMDMs treated with LPS for 4 hours, with or without OI pretreatment (0.25 mM, 2 hours). Significance of qPCR was analyzed with one-way ANOVA corrected with Bonferroni's post hoc test for multiple comparisons, *n* = 3; *p<0.05, **p<0.01, ***p<0.001.

The online version of this article includes the following figure supplement(s) for figure 9:

**Figure supplement 1.** NRF2 is not required for the anti-inflammatory effects of exogenous itaconate (OI) on LPS-induced inflammatory response.

(*Figure 10A*). Inflammatory cytokine genes (*Il6*, *Tnfa*, and *Il1b*) were significantly induced by PM in AMs from both WT and *Acod1*[-/-] mice. There was no difference in levels of inflammatory cytokine genes between the WT and *Acod1*[-/-] groups (*Figure 10B*). Analysis of cytokine levels in the BAL fluid by ELISA showed similar increases in IL-6 and TNFα following PM treatment, with no differences between WT and *Acod1*[-/-] mice (*Figure 10C*). Collectively, these data suggest that itaconate does not play a role in PM-induced inflammation in vitro or in vivo.

## Discussion

Exposure to PM air pollution is associated with significant morbidity and mortality (*Hamanaka and Mutlu, 2018*). Air pollution exposure is one of the top preventable causes of death in the world (*McGlade and Landrigan, 2019*). It is estimated that exposure to air pollution causes 4.2 million premature deaths worldwide every year according to the World Health Organization (*Hamanaka and Mutlu, 2018*; *WHO, 2018*). Since limiting exposure to PM cannot be achieved immediately, a better

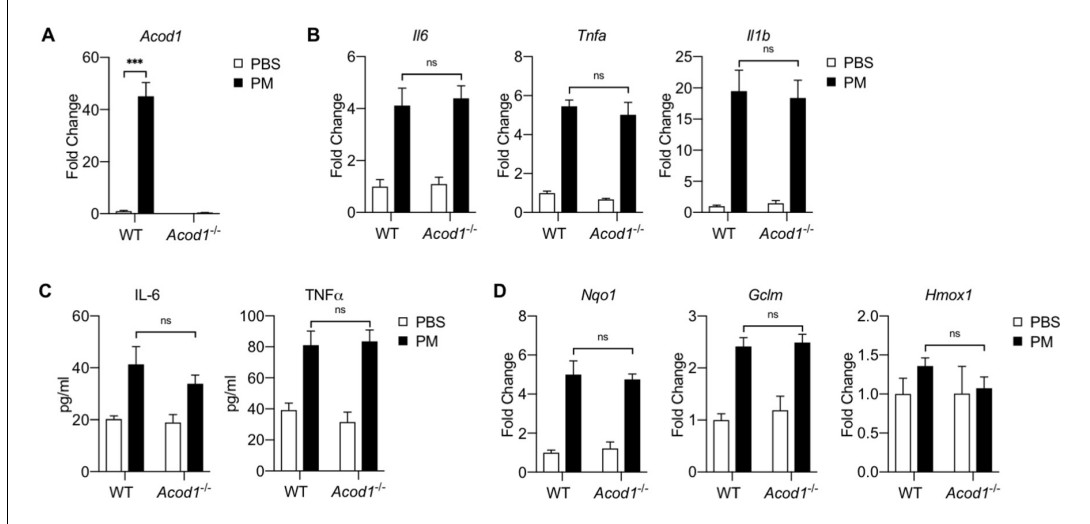

**Figure 10.** Acod1 and Itaconate are not required for PM-induced inflammation in an in vivo model of PM exposure. WT and *Acod1⁻/⁻*mice were treated intratracheally with PM (100 µg) or PBS and 24 hours later, bronchoalveolar lavage (BAL) fluid was collected to obtain alveolar macrophages and cytokine levels in fluid. (**A**) *Acod1* gene expression is induced in alveolar macrophages by in vivo PM treatment in WT mice as measured by qPCR. (**B**) Inflammatory genes (*Il6, Tnfa, Il1b*) as measured by qPCR were equally upregulated by PM treatment in vivo in both WT and *Acod1⁻/⁻* mice. (**C**) Cytokine levels of IL-6 and TNFα in BAL fluid as measured by ELISA; IL-6 and TNFα levels increase equally with PM treatment in both WT and *Acod1⁻/⁻* mice. (**D**) NRF2 target genes (*Nqo1, Gclm, Hmox1*) were upregulated with PM treatment as measured by qPCR, with no difference between WT and *Acod1⁻/⁻* mice. Significance was analyzed with one-way ANOVA corrected with Bonferroni's post hoc test for multiple comparisons, n = 6; *p<0.05, **p<0.01, ***p<0.001.

The online version of this article includes the following figure supplement(s) for figure 10:

**Figure supplement 1.** Acod1 and Itaconate are not required for PM-induced inflammatory response in tissue resident alveolar macrophages.

understanding of the mechanisms by which PM causes morbidity and mortality is required in order to combat the effects of air pollution on public health.

As lungs are the entry site for PM, it is not surprising that lung inflammation plays an important role in PM-induced adverse health effects, which are predominantly cardiopulmonary diseases. We have previously reported that macrophages, and in particular, their production of IL-6, are required for the PM-induced lung inflammation and resultant acute thrombotic cardiovascular events (*Mutlu et al., 2007*; *Chiarella et al., 2014*; *Soberanes et al., 2019*). However, the mechanisms regulating PM-induced inflammation in the lung are not completely understood.

In this study, using an unbiased approach with RNA-seq, we first discovered *Acod1* as the gene most induced by PM in macrophages. In a time-dependent fashion, PM induced the mRNA and protein expression of ACOD1, which was not evident at earlier time points (4, 8 hours) but was detected at 24 hours. Confirming a functional role, PM-induced expression of ACOD1 led to production of itaconate in our metabolomics measurements.

Assessment of the effect of PM on mitochondrial function showed that PM has a time-dependent effect on mitochondrial respiration. While PM increased mitochondrial OCR at 1 hour, it reduced OCR at 24 hours. Similarities in the time dependency between the effects of PM on ACOD1/itaconate and mitochondrial respiration led us to explore whether itaconate is responsible for the late reduction of OCR following PM exposure. Using *Acod1⁻/⁻* macrophages, we found that itaconate is required for the PM-induced reduction in mitochondrial OCR at 24 hours. Consistent with a role of itaconate as an inhibitor of succinate dehydrogenase, we found that *Acod1⁻/⁻* BMDMs lack succinate accumulation following PM treatment. This finding is in agreement with findings by Cordes et al., which showed a loss of succinate accumulation in response to LPS stimulation in *Acod1⁻/⁻* macrophages (*Cordes et al., 2016*). Collectively, our data support a key role for Acod1 and itaconate in the metabolic reprogramming of macrophages in response to both LPS and PM stimulation.

Interestingly, although we found that endogenous itaconate production was an important regulator of mitochondrial metabolism and respiration in PM-treated macrophages, we were unable to

find an effect of endogenous itaconate on PM-induced inflammation. This was in contrast to the effect of *Acod1* deletion on LPS-induced inflammation, in which IL-6 and IL-1β induction was augmented in the absence of endogenous itaconate production. Our results show a stimulus dependency for the effect of itaconate that has not been previously demonstrated. Thus, although PM and LPS both promote inflammatory cytokine expression and itaconate-dependent metabolic reprogramming, the effects of itaconate on inflammation differ between the two stimuli.

To enhance our understanding of the role of itaconate in the lungs with PM exposure, we used an in vivo model of PM exposure in WT and *Acod1*⁻/⁻ mice. Similar to our in vitro data, in vivo exposure of mice to PM shows that while *Acod1* is highly upregulated in the lung macrophages following PM exposure, the presence or lack of *Acod1* did not affect inflammation in the lung. Expression of IL-6 and TNFα were equally increased with PM exposure, regardless of ACOD1 expression status.

To date, the majority of studies investigating the role of Acod1 and itaconate on metabolism have largely focused on the effects of exogenously-applied cell membrane-permeable forms of itaconate (e.g., dimethyl itaconate and OI) (*Lampropoulou et al., 2016*; *Bambouskova et al., 2018*; *Zhao et al., 2019*). Following the initial studies with dimethyl itaconate (*Lampropoulou et al., 2016*; *Bambouskova et al., 2018*; *Zhao et al., 2019*), a more recent study questioned whether treatment with dimethyl itaconate actually increases intracellular levels of itaconate (*ElAzzouny et al., 2017*). In response to this limitation of dimethyl itaconate, Mills and colleagues proposed the use of OI as a cell permeable form of itaconate and demonstrated that OI does increase intracellular levels of itaconate (*Mills et al., 2018*). Our results showing that OI reduces PM-induced cytokine production are in agreement with Mills and colleagues suggesting that the exogenous cell membrane-permeable version of itaconate has anti-inflammatory effects. However, given the differing effects of endogenous itaconate production versus OI treatment on PM-treated macrophages, our results suggest additional cautions should be taken when extrapolating the effects of cell-permeable itaconate analogs to the effects of endogenously-produced itaconate. This is highlighted by the fact that while OI is sufficient to induce NRF2 activation in macrophages, Acod1 is not required for NRF2 activation in response to PM or LPS. It would be expected that if a major effect of endogenous itaconate production were NRF2 activation, this should have been absent in *Acod1*⁻/⁻ BMDMs. This loss of function experiment was notably lacking in the previous report linking itaconate with NRF2. Our results suggest that location of itaconate plays a major role in regulating its downstream effects.

While our results agree with Mills et al that OI activates the NRF2 antioxidant pathway by upregulation of NRF2 protein and its target genes, we found that NRF2 was dispensable for the anti-inflammatory effects of OI. Mills et al assessed the anti-inflammatory effects of OI by Western blot for intracellular IL-1β. Our findings examining the expression of multiple cytokines at the mRNA level suggest that NRF2 does not play a major role in regulating the anti-inflammatory effect of OI.

In conclusion, we found that exposure of macrophages to PM induces a mitochondrial enzyme, ACOD1 and production of a mitochondrial metabolite, itaconate, which is required for PM-induced metabolic reprogramming. Although endogenous itaconate reprograms mitochondrial function in PM-treated macrophages, it does not have a major effect on PM-induced gene expression or inflammatory response in vitro or in vivo. Future work will be required to determine whether endogenous itaconate regulates other macrophage functions. Our finding that itaconate is secreted from macrophages suggests that important, unexplored effects of itaconate production may occur in non-macrophage cells. Our results also caution about the interpretation of results achieved using exogenous cell membrane-permeable forms of itaconate as they may not represent the effects of endogenously-produced itaconate. Further, our results suggest that our understanding of the role of NRF2 in the cellular response to itaconate is far from complete. While OI is sufficient to induce NRF2 activation in macrophages, endogenous itaconate production is not required for PM or LPS-mediated NRF2 activation.

# Materials and methods

## Key resources table

| Reagent type (species) or resource | Designation | Source or reference | Identifiers | Additional information |
|---|---|---|---|---|
| Gene (*M. musculus*) | Acod1 | | Aconitate decarboxylase 1; Irg1 | |
| Strain, strain background (*M. musculus*) | C57BL/6NJ (WT) | Jackson Laboratory | Stock No. 005304 | Male (7–11 weeks) |
| Strain, strain background (*M. musculus*) | C57BL/6NJ-*Acod1^em1(IMPC)J*/J (Acod1 KO) | Jackson Laboratory | Stock No. 029340, RRID:IMSR_JAX:029340 | Male (7–11 weeks) |
| Transfected construct (*M. musculus*) | Non-targeting siRNA | Dharmacon | D-001810-01-05 | |
| Transfected construct (*M. musculus*) | *Nfe2l2* siRNA #1 | Dharmacon | J-040766-08-0002 | |
| Transfected construct (*M. musculus*) | *Nfe2l2* siRNA #2 | Dharmacon | J-040766-06-0002 | |
| Antibody | Anti-β-actin (mouse, monoclonal) | Sigma Aldrich | Catalog #: A5441, RRID:AB_476744 | WB (1:10,000) |
| Antibody | Anti-ACOD1 (rabbit, polyclonal) | Thermo Fisher Scientific | Catalog #: PA5-49094, RRID:AB_2634550 | WB (1:500) |
| Antibody | Anti-NRF2 (rabbit, monoclonal) | Abcam | Catalog #: Ab62352, RRID:AB_944418 | WB (1:1000) |
| Antibody | Anti-iNOS (rabbit, polyclonal) | Cell Signaling Technology | Catalog #: 2982, RRID:AB_1078202 | WB (1:1000) |
| Antibody | Anti-rabbit IgG, HRP-linked Antibody | Cell Signaling Technology | Catalog #: 7074, RRID:AB_2099233 | WB (1:2000) |
| Antibody | Anti-mouse IgG, HRP-linked Antibody | Cell Signaling Technology | Catalog #: 7076, RRID:AB_330924 | WB (1:2000) |
| Sequence-based reagent | RPL13a_F | This paper | PCR Primers | GAGGTCGGGTGGAAGTACCA |
| Sequence-based reagent | RPL13a_R | This paper | PCR Primers | TGCATCTTGGCCTTTTCCTT |
| Sequence-based reagent | Acod1_F | This paper | PCR Primers | TTTGGGGTCGACCAGACTTC |
| Sequence-based reagent | Acod1_R | This paper | PCR Primers | CCATGGAGTGAACAGCAACAC |
| Sequence-based reagent | Il6_F | This paper | PCR Primers | TCCTCTCTGCAAGAGACTTCC |
| Sequence-based reagent | Il6_R | This paper | PCR Primers | AGTCTCCTCTCCGGACTTGT |
| Sequence-based reagent | Tnfa_F | This paper | PCR Primers | ATGGCCTCCCTCTCATCAGT |
| Sequence-based reagent | Tnfa_R | This paper | PCR Primers | TGGTTTGCTACGACGTGGG |
| Sequence-based reagent | Il1b_F | This paper | PCR Primers | GCCACCTTTTGACAGTGATGA |
| Sequence-based reagent | Il1b_R | This paper | PCR Primers | GACAGCCCAGGTCAAAGGTT |
| Sequence-based reagent | Nqo1_F | This paper | PCR Primers | GGTAGCGGCTCCATGTACTC |

*Continued on next page*

*Continued*

| Reagent type (species) or resource | Designation | Source or reference | Identifiers | Additional information |
|---|---|---|---|---|
| Sequence-based reagent | Nqo1_R | This paper | PCR Primers | CGCAGGATGCCACTCTGAAT |
| Sequence-based reagent | Gclm_F | This paper | PCR Primers | AGTTGACATGGCATGCTCCG |
| Sequence-based reagent | Gclm_R | This paper | PCR Primers | CCATCTTCAATCGGAGGCGA |
| Sequence-based reagent | Hmox1_F | This paper | PCR Primers | GAGCAGAACCAGCCTGAACT |
| Sequence-based reagent | Hmox1_R | This paper | PCR Primers | AAATCCTGGGGCATGCTGTC |
| Sequence-based reagent | Nos2_F | This paper | PCR Primers | TTCACAGCTCATCCGGTACG |
| Sequence-based reagent | Nos2_R | This paper | PCR Primers | TCGATGCACAACTGGGTGAA |
| Commercial assay or kit | Direct-zol RNA Miniprep Kits | Zymo Research | R2053 | |
| Commercial assay or kit | Mouse IL-6 DuoSet ELISA | R and D Systems | DY406 | |
| Commercial assay or kit | Mouse TNF-alpha DuoSet ELISA | R and D Systems | DY410 | |
| Commercial assay or kit | Seahorse XFe24 FluxPak | Agilent | 102340–100 | |
| Commercial assay or kit | Seahorse XF Plasma Membrane Permeabilizer | Agilent | 102504–100 | |
| Commercial assay or kit | Mouse Macrophage Nucleofector Kit | Lonza | VPA-1009 | |
| Chemical compound, drug | 4-Octyl Itaconate | Sigma Aldrich | SML2338 | |
| Chemical compound, drug | Itaconic Acid | Sigma Aldrich | I29204 | |
| Chemical compound, drug | Malonic Acid | Sigma Aldrich | M1296 | |
| Chemical compound, drug | Pyruvic Acid | Sigma Aldrich | 107360 | |
| Chemical compound, drug | Malic Acid | Sigma Aldrich | 02288 | |
| Chemical compound, drug | Succinic Acid | Sigma Aldrich | S3674 | |
| Chemical compound, drug | Particulate Matter (PM) | NIST | Urban Dust - SRM 1649a | |
| Chemical compound, drug | Lipopolysaccharide | Santa Cruz | sc-3535 | |
| Chemical compound, drug | Oligomycin | Fisher Scientific | 49-545-510MG | |
| Chemical compound, drug | FCCP | Sigma Aldrich | C2920 | |
| Chemical compound, drug | Antimycin A | Sigma Aldrich | A8674 | |
| Chemical compound, drug | Rotenone | Sigma Aldrich | R8875 | |
| Chemical compound, drug | Recombinant Mouse M-CSF | BioLegend | 576408 | |

*Continued on next page*

*Continued*

| Reagent type (species) or resource | Designation | Source or reference | Identifiers | Additional information |
|---|---|---|---|---|
| Software, algorithm | Prism 8 | GraphPad | RRID:SCR_002798 | |
| Software, algorithm | FastQC | Babraham Institute | RRID:SCR_014583 | |
| Software, algorithm | STAR | PMID:23104886 | RRID:SCR_015899 | |
| Software, algorithm | Picard | Broad Institute | RRID:SCR_006525 | |
| Software, algorithm | RSeQC | PMID:22743226 | RRID:SCR_005275 | |
| Software, algorithm | FeatureCounts | WEHI | RRID:SCR_012919 | |
| Software, algorithm | DESeq2 | Bioconductor | RRID:SCR_015687 | |

## Cell isolation and culture

All animal experiments and procedures were performed according to the protocols approved by the Institutional Animal Care and Use Committee at the University of Chicago. We used primary murine cells (bone marrow-derived macrophages (BMDMs)), which we isolated as we have recently reported (*Woods et al., 2020*). Hematopoietic cells were isolated from bone marrow of C57BL/6NJ (Stock No: 005304) and *Acod1*$^{-/-}$ (Stock No: 029340) (both from Jackson Laboratory) mice and cultured with M-CSF (20 µg/L, BioLegend, catalog number 576408) in vitro for 8–10 days to generate BMDMs. Tissue-resident alveolar macrophages were isolated by standard BAL fluid collection, as we have previously described (*Chiarella et al., 2014*; *Soberanes et al., 2019*), and cultured in media for 2 hr before treatment. For all experiments, cells were cultured in complete medium containing RPMI (Gibco, catalog number A10491), supplemented with 10% heat-inactivated FBS (Gemini, catalog number 100–106) and 1% penicillin-streptomycin (Gemini, 400–109).

Reagents were purchased from Sigma-Aldrich, including: 4-Octyl itaconate (catalog number SML2338), Itaconic Acid (catalog number I29204), Malonic Acid (M1296). Particulate matter (SRM 1649a, Urban Dust) was from National Institute of Standards Technology (NIST). Lipopolysaccharide was purchased from Santa Cruz (sc-3535). Full details of reagents used are included in the Key Resources Table.

## Cell lysis and western blotting

Cells were scraped into RIPA buffer (Thermo-Scientific, catalog number 89900) with protease and phosphatase inhibitors (Thermo-Scientific, 1861284), sonicated for 10 s on a Fisher Scientific 100 model at speed setting 2. Samples were resolved by SDS-PAGE on 10% polyacrylamide gels and transferred to nitrocellulose (Bio-Rad, catalog number 1620167). Primary antibodies used were mouse anti-β-actin monoclonal antibody (Sigma, catalog number A5441; lot number 037K488; 1:10,000), ACOD1 polyclonal antibody (Invitrogen, catalog number PA5-49094, 1:500), NRF2 monoclonal antibody (Abcam, catalog number ab62352, 1:1000), iNOS Antibody (Cell Signaling Technology, catalog number 2982S, 1:1000). Secondary antibodies used were anti-rabbit IgG HRP-linked antibody (Cell Signaling Technology, catalog number 7074S) and anti-mouse IgG HRP-linked antibody (Cell Signaling Technology, catalog number 7076S).

## Quantitative PCR

Total RNA was extracted with TRI Reagent (Zymo Research, R2050-1-200). RNA was then isolated using the Zymo Direct-zol RNA Miniprep Kit (Zymo Research, catalog number R2053) and reverse-transcribed using Bio-Rad iScript Reverse Transcription Supermix (Bio-Rad, catalog number 1708841) in a Bio-Rad C1000 Touch Thermal Cycler. Quantitative mRNA expression was determined by real-time qPCR using iTaq Universal SYBR Green Supermix (Bio-Rad, catalog number 172–5121). qPCR primer sequences used are as follows:

*Rpl13a* (control) (5'-GAGGTCGGGTGGAAGTACCA-3', 5'-TGCATCTTGGCCTTTTCCTT-3'),
*Acod1* (5'-TTTGGGGTCGACCAGACTTC-3', 5'-CCATGGAGTGAACAGCAACAC-3'),
*Il6* (5'-TCCTCTCTGCAAGAGACTTCC-3', 5'-AGTCTCCTCTCCGGACTTGT-3'),
*Tnfa* (5'-ATGGCCTCCCTCTCATCAGT-3', 5'-TGGTTTGCTACGACGTGGG-3'),
*Il1b* (5'-GCCACCTTTTGACAGTGATGA-3',5'-GACAGCCCAGGTCAAAGGTT-3'),
*Nqo1* (5'-GGTAGCGGCTCCATGTACTC-3', 5'-CGCAGGATGCCACTCTGAAT-3'),
*Gclm* (5'-AGTTGACATGGCATGCTCCG-3', 5'-CCATCTTCAATCGGAGGCGA-3'),
*Hmox1* (5'-GAGCAGAACCAGCCTGAACT −3', 5'- AAATCCTGGGGCATGCTGTC-3'),
*Nos2* (5'-TTCACAGCTCATCCGGTACG −3', 5'- TCGATGCACAACTGGGTGAA-3').

## ELISA

Cells were treated in complete medium for 24 hours, and the media was collected. IL-6 and TNFα cytokine levels in the media were then measured with DuoSet ELISA kits (R and D systems, catalog number DY406 and DY410) according to manufacturer's protocol.

## RNA sequencing

RNA was isolated and submitted for sequencing (50 bp SE). Quality assessment of sequencing files was assessed with FastQC. Reads were mapped to the GRCm38 reference genome using STAR aligner (*Dobin et al., 2013*), and quality was assessed using Picard tools and RSeQC (*Wang et al., 2012*). Genes were quantified using featureCounts, and low expression genes were removed at cpm = 1.5. Differentially expressed genes (DEGs) were identified using DESeq2 analysis at FC > 2 and FDR adjusted p-value p<0.05.

## Metabolomics

BMDMs were plated at $3 \times 10^6$ cells on 60 mm tissue culture plates for 2 hours, then treated with PM (20 μg/cm$^2$) for 24 hours prior to metabolite extraction. Cells were washed with 5% mannitol solution and metabolites were extracted with 400 μl methanol. 275 μl internal standard was added, then the extracts were centrifuged at $2,300 \times g$ for 5 min. The supernatant was transferred to pre-washed centrifugal filter units (HMT, Human Metabolome Technologies, Boston, MA) and centrifuged at $9,100 \times g$ at 4°C for 2 hours. Centrifuged samples were sent to HMT for processing and analysis.

## Extracellular itaconate measurement

This measurement was performed by Mass spectrometry, Metabolomics and Proteomics Facility in Research Resources Center of University of Illinois at Chicago. All samples are analyzed by Waters ACQUITY UPLC BEH C18 Column, 130A°, 1.7 μm, 2.1 mm X 100 mm coupled to an Agilent 1260 UPLC system, which was operated at a flow rate of 500 uL/min. A linear gradient of 1–60% buffer B (100% MeOH) was applied. MS data were acquired by MRM scan (Negative ESI spray voltage 4.5kV, temperature 550 degrees, m/z range 50–300) monitoring signature product ions 129 > 69 (Quantifier) and 129 > 59 (Qualifier) transitions. The quantification was achieved using peak area of monitored transitions. All the samples were analyzed by triplicate.

## siRNA knockdowns

BMDMs ($1 \times 10^6$ cells) were transfected with siRNA (250 pmol) and Amaxa Mouse Macrophage Nucleofector Kit (Lonza, catalog number VPA-1009) using a Lonza Nucleofector 2b device (Lonza, #AAB-1001) on the mouse macrophage (Y-001) setting. Cells were cultured for 2 days post transfection before treatment. Success of siRNA transfections were confirmed with western blots and qPCR. siRNAs were purchased from Dharmacon: D-001810-01-05 (non-targeting siRNA); J-040766-08-0002 (*Nfe2l2* #1), J-040766-06-0002 (*Nfe2l2* #2).

## Seahorse analysis

The Seahorse XF$^e$24 Extracellular Flux Analyzer (Seahorse Bioscience, North Billerica, MA), was used to measure oxygen consumption rates (OCR) and extracellular acidification rates (ECAR) as we have previously described (*Nigdelioglu et al., 2016*; *Hamanaka et al., 2019*; *Soberanes et al., 2019*;

*Woods et al., 2020*). Macrophages were seeded at a density of $4 \times 10^4$ per well on a Seahorse XF24 Cell Culture Microplate (*Woods et al., 2020*).

Mitochondrial stress tests were performed according to manufacturer's protocol in XF DMEM base medium (Agilent 103334–100) containing Glutamine (2 mM), Sodium Pyruvate (1 mM), and Glucose (25 mM). Compounds of interest were injected, followed by sequential injections of Oligomycin (1.5 µM), FCCP (1.5 µM) and Antimycin A/Rotenone (1.25 µM).

Membrane permeabilization assays were performed in 1x MAS buffer containing mannitol (220 mM), sucrose (70 mM), monopotassium phosphate (10 mM), magnesium chloride (5 mM), HEPES (2 mM), EGTA (1 mM), with the addition of Seahorse XF Plasma Membrane Permeabilizer (Agilent, catalog number 102504–100) and Fatty Acid Free BSA (0.2%). Substrates and reagents used in permeabilization assays include succinate (10 mM), rotenone (2 mM), pyruvate (10 mM), malate (0.5 mM), ADP (4 mM), itaconate (10 mM), malonate (10 mM), oligomycin (2 mM), antimycin A (2 mM). All substrates and reagents were pH-adjusted to 7.2 prior to assay with 5M potassium hydroxide (KOH) solution. Permeabilization assays were performed with cycles of 0.5 min mix, 0.5 min wait, and 2 min measure intervals, as per manufacturer's protocol.

## In vivo mouse studies

C57BL/6NJ and *Acod1*[-/-] mice were treated via intratracheal instillation with only PBS or PM in PBS as we have previously described (*Mutlu et al., 2006*; *Mutlu et al., 2007*). PM was freshly diluted to a 2.5 mg/ml stock, and 2 doses of 20 µl PM stock was given intratracheally to each mouse, for a total of 100 µg in 40 µl PBS per mouse. For the PBS controls, 2 doses of 20 µl PBS were given intratracheally to each mouse. Following 24 hours of exposure, BAL fluid was obtained by flushing the lungs with 0.5 mM EDTA in ice cold PBS as we have previously described (*Chiarella et al., 2014*; *Soberanes et al., 2019*). The first wash of 500 µl was collected and centrifuged, where the supernatant was then used for ELISA studies. The cell pellet was combined with the cells obtained from the subsequent eight washes, and used for analysis of mRNA expression through qPCR analysis.

## Statistical analysis

Data were analyzed using Prism 8 (GraphPad Software, Inc). All data are shown as mean ± standard error of the mean (SEM). Significance was determined by unpaired two-tailed Student's t test (for comparisons between two samples), or by one-way ANOVA using Bonferroni correction for multiple comparisons. $*p<0.05$, $**p<0.01$, $***p<0.001$.

## Acknowledgements

**Funding:** T32HL007605 (PSW, and LMK), K01AR066579, ATS Foundation Grant, and Respiratory Health Association Grant RHA2018-01-IPF (RBH) and R01ES010524, U01ES026718, P01HL144454, P30ES027792 and Department of Defense W81XWH-16-1-0711 (GMM).

## Additional information

### Funding

| Funder | Grant reference number | Author |
| --- | --- | --- |
| National Institute of Environmental Health Sciences | R01ES010524 | Gökhan M Mutlu |
| National Institute of Environmental Health Sciences | U01ES026718 | Gökhan M Mutlu |
| National Institute of Environmental Health Sciences | P30ES027792 | Gökhan M Mutlu |
| National Heart, Lung, and Blood Institute | P01HL144454 | Gökhan M Mutlu |
| National Heart, Lung, and Blood Institute | T32HL007605 | Parker S Woods Lucas M Kimmig |

| National Institute of Arthritis and Musculoskeletal and Skin Diseases | K01AR066579 | Robert B Hamanaka |
|---|---|---|
| American Thoracic Society | Unrestricted Grant | Robert B Hamanaka |
| Respiratory Health Association | RHA2018-01-IPF | Robert B Hamanaka |
| U.S. Department of Defense | W81XWH-16-1-0711 | Gökhan M Mutlu |

The funders had no role in study design, data collection and interpretation, or the decision to submit the work for publication.

## Author contributions

Kaitlyn A Sun, Conceptualization, Formal analysis, Investigation, Methodology, Writing - original draft, Writing - review and editing; Yan Li, Data curation, Formal analysis, Validation, Investigation, Methodology, Writing - review and editing; Angelo Y Meliton, Data curation, Formal analysis, Supervision, Methodology, Writing - review and editing; Parker S Woods, Data curation, Formal analysis, Investigation, Methodology, Writing - review and editing; Lucas M Kimmig, Data curation, Formal analysis, Investigation; Rengül Cetin-Atalay, Data curation, Formal analysis, Writing - review and editing; Robert B Hamanaka, Formal analysis, Supervision, Funding acquisition, Investigation, Writing - original draft, Project administration, Writing - review and editing; Gökhan M Mutlu, Conceptualization, Resources, Formal analysis, Supervision, Investigation, Writing - original draft, Project administration, Writing - review and editing

## Author ORCIDs

Robert B Hamanaka  http://orcid.org/0000-0002-8909-356X
Gökhan M Mutlu  https://orcid.org/0000-0002-2056-612X

## Ethics

Animal experimentation: This study was performed in strict accordance with the recommendations in the Guide for the Care and Use of Laboratory Animals of the National Institutes of Health. All animals were handled according to approved institutional animal care and use committee (IACUC) protocols (72376 and 72465) of the University of Chicago.

## Decision letter and Author response

Decision letter https://doi.org/10.7554/eLife.54877.sa1
Author response https://doi.org/10.7554/eLife.54877.sa2

# Additional files

## Supplementary files

- Source code 1. R code for differential expression analysis (DESeq2).
- Source code 2. R code for differential expression analysis.
- Transparent reporting form

## Data availability

Sequencing data have been deposited in GEO under accession code GSE143881. In addition, source data files have been provided for Figure 1, Figure 6A, 6B and 6C.

The following dataset was generated:

| Author(s) | Year | Dataset title | Dataset URL | Database and Identifier |
|---|---|---|---|---|
| Li Y, Sun KA, Mutlu GM | 2020 | Endogenous itaconate is not required for particulate matter-induced NRF2 expression or inflammatory response | https://www.ncbi.nlm.nih.gov/geo/query/acc.cgi?acc=GSE143881 | NCBI Gene Expression Omnibus, GSE143881 |

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
