## [Decision Letter]

**Acceptance summary:**

The metabolite itaconate has recently gained a lot of interest for its anti-inflammatory role in inflammatory macrophages. Here Mutlu and colleagues found that endogenous itaconate produced by the enzyme Acod1 does not influence inflammatory responses in macrophages exposed to particulate matter (PM) in vitro or in a mouse model of PM-induced lung inflammation, or to the microbial product LPS.

**Decision letter after peer review:**

Thank you for submitting your article "Endogenous itaconate is not required for particulate matter-induced NRF2 expression or inflammatory response" for consideration by *eLife*. Your article has been reviewed by three peer reviewers, one of whom is a member of our Board of Reviewing Editors, and the evaluation has been overseen by Satyajit Rath as the Senior Editor. The following individuals involved in review of your submission have agreed to reveal their identity: Jan Van den Bossche (Reviewer #2); Daniel McVicar (Reviewer #3).

The reviewers have discussed the reviews with one another and the Reviewing Editor has drafted this decision to help you prepare a revised submission.

Summary:

Particulate matter from air pollution is known to induce macrophage inflammation, and in this manuscript, Mutlu and colleagues investigated the role of ACOD1/itaconate. The authors found that macrophages exposed to PM upregulated Acod1 and production of itaconate. Such itaconate production contributed to the ability of PM to suppress oxidative metabolism via effects on Complex II, although it did not appear to affect inflammatory gene induction. The authors also found that endogenous and exogenous itaconate have different effects on the macrophage response to PM, which is of interest given the many recent studies that have explored the use of OI and dimethyl-itaconate (DI) as therapeutics. The authors also found that endogenous itaconate (via ACOD1) and exogenous itaconate do not regulate NRF2 to modulate anti-inflammatory effects, in macrophages stimulated with either PM or bacterial LPS.

Essential revisions:

1) Although the objective of this study was to understand the mechanisms behind PM-induced lung inflammation, the authors relied exclusively on PM stimulation of BMDMs in vitro and did not see any effect of ACOD1-mediated itaconate production on inflammatory gene induction. To strengthen physiological relevance, the authors should use models of PM-induced lung inflammation and examine lung and alveolar macrophages for induction of ACOD1 and production of itaconate, and compare WT and Acod1 KO mice for lung inflammation to determine the role of itaconate (or if not possible, co-administration of OI in WT mice).

2) The authors should provide more details in Materials and methods (e.g., concentration of itaconate used) so that readers can better understand what was done and to allow for experiments to be reproduced by the scientific community.

---

## [Author Response]

Essential revisions:1) Although the objective of this study was to understand the mechanisms behind PM-induced lung inflammation, the authors relied exclusively on PM stimulation of BMDMs in vitro and did not see any effect of ACOD1-mediated itaconate production on inflammatory gene induction. To strengthen physiological relevance, the authors should use models of PM-induced lung inflammation and examine lung and alveolar macrophages for induction of ACOD1 and production of itaconate, and compare WT and Acod1 KO mice for lung inflammation to determine the role of itaconate (or if not possible, co-administration of OI in WT mice).

We agree that an in vivo model would enhance the physiological relevance of this study. We treated WT and *Acod1^-/-^* mice with either PBS or PM intratracheally and similarly found that itaconate does not affect inflammatory gene or cytokine production. Details of this experiment can be found in Figure 10. Furthermore, we found that in tissue resident alveolar macrophages, loss of Acod1 did not affect PM-induced proinflammatory response or NRF2 activation (Figure 10—figure supplement 1).

2) The authors should provide more details in Materials and methods (e.g., concentration of itaconate used) so that readers can better understand what was done and to allow for experiments to be reproduced by the scientific community.

As suggested, we have revised the Materials and methods to include specific details, including concentrations and treatment times, for more clarity.